# ERβ Regulation of Indian Hedgehog Expression in the First Wave of Ovarian Follicles

**DOI:** 10.3390/cells13070644

**Published:** 2024-04-06

**Authors:** V. Praveen Chakravarthi, Iman Dilower, Subhra Ghosh, Shaon Borosha, Ryan Mohamadi, Vinesh Dahiya, Kevin Vo, Eun B. Lee, Anamika Ratri, Vishnu Kumar, Courtney A. Marsh, Patrick E. Fields, M. A. Karim Rumi

**Affiliations:** 1Department of Pathology and Laboratory Medicine, University of Kansas Medical Center (KUMC), Kansas City, KS 66160, USA; praghavulu@kumc.edu (V.P.C.); idilower2@kumc.edu (I.D.); subhrazoology@gmail.com (S.G.); sbrcb@missouri.edu (S.B.); r346m406@ku.edu (R.M.); vinesh.dahiyampharm@gmail.com (V.D.); vokevin816@gmail.com (K.V.); elee9871@gmail.com (E.B.L.); aratri@kumc.edu (A.R.); vak39@case.edu (V.K.); pfields@kumc.edu (P.E.F.); 2Obstetrics and Gynecology, University of Kansas Medical Center (KUMC), Kansas City, KS 66160, USA; cmarsh2@kumc.edu

**Keywords:** ovarian development, primordial follicle activation, estrogen receptor β, transcriptome analysis, Indian hedgehog, the first wave of ovarian follicles, the second wave of ovarian follicles

## Abstract

Increased activation of ovarian primordial follicles in *Erβ* knockout (*Erβ^KO^*) rats becomes evident as early as postnatal day 8.5. To identify the ERβ-regulated genes that may control ovarian primordial follicle activation, we analyzed the transcriptome profiles of *Erβ^KO^* rat ovaries collected on postnatal days 4.5, 6.5, and 8.5. Compared to wildtype ovaries, *Erβ^KO^* ovaries displayed dramatic downregulation of Indian hedgehog (*Ihh*) expression. IHH-regulated genes, including *Hhip*, *Gli1,* and *Ptch1,* were also downregulated in *Erβ^KO^* ovaries. This was associated with a downregulation of steroidogenic enzymes *Cyp11a1*, *Cyp19a1*, and *Hsd17b1*. The expression of *Ihh* remained very low in *Erβ^KO^* ovaries despite the high levels of *Gdf9* and *Bmp15*, which are known upregulators of *Ihh* expression in the granulosa cells of activated ovarian follicles. Strikingly, the downregulation of the *Ihh* gene in *Erβ^KO^* ovaries began to disappear on postnatal day 16.5 and recovered on postnatal day 21.5. In rat ovaries, the first wave of primordial follicles is rapidly activated after their formation, whereas the second wave of primordial follicles remains dormant in the ovarian cortex and slowly starts activating after postnatal day 12.5. We localized the expression of *Ihh* mRNA in postnatal day 8.5 wildtype rat ovaries but not in the age-matched *Erβ^KO^* ovaries. In postnatal day 21.5 *Erβ^KO^* rat ovaries, we detected *Ihh* mRNA mainly in the activated follicles in the ovaries’ peripheral regions. Our findings indicate that the expression of *Ihh* in the granulosa cells of the activated first wave of ovarian follicles depends on ERβ.

## 1. Introduction

Primordial follicle assembly following oocyte nest breakdown is the first step in ovarian folliculogenesis [1]. In mice and rats, primordial follicles develop in two distinct waves [2,3]. The first wave of primordial follicles is formed in the ovarian medulla during the first three days after birth and is rapidly activated after formation [2,3]. In contrast, the second wave of primordial follicles is assembled in the ovarian cortex during postnatal days 4.5 to 7.5, and most of these primordial follicles remain dormant and serve as the ovarian reserve [2,3]. A few second wave primordial follicles are selectively activated through a highly regulated intraovarian mechanism known as primordial follicle activation. The second wave of primordial follicles undergoes primordial follicle activation starting from postnatal day 12.5 and gradually replenishes the loss of activated first wave follicles [2,3]. The first wave follicles are completely depleted in three months, and the second wave follicles perform the ovarian functions [2,3]. Thus, the regulation of primordial follicle activation is a crucial mechanism for maintaining ovarian reserves and the female reproductive lifespan [4]. However, the role of the rapidly activated and short-lived first wave of primordial follicles in ovarian biology remains unknown [5]. It is suggested to play a role in the establishment of early reproductive functions and the hypothalamic–pituitary axis of endocrine regulation of ovarian functions. Recently, it has been reported that XO female mice, a model of Turner syndrome, lack the first wave of primordial follicles [6]. Further follow-up of XO mice has shown that they develop premature ovarian insufficiency [7]. These findings suggest that the first wave of ovarian follicles may play an important role in maintaining the second wave of primordial follicle reserve [7]. 

Estrogen receptor β (ERβ) is the predominant estrogen receptor in the mammalian ovary and is essential for follicle development and ovulation [8,9,10]. We previously identified that ERβ plays an important role in controlling the normal rate of primordial follicle activation [11]. Loss of ERβ increased primordial follicle activation, leading to premature ovarian senescence in *Erβ* knockout (*Erβ^KO^*) rats [11]. Regulation of primordial follicle activation is a complex mechanism involving several negative regulators within the PI3-kinase and mTOR pathways [12]. Loss of ERβ was associated with increased activation of AKT, ERK, and mTOR pathways in *Erβ^KO^* ovaries [11]. ERβ may also inhibit the induction of primordial follicle activation like the transcriptional regulators FOXO3A and FOXL2 [11,12,13]. However, the precise mechanism of the ERβ-mediated regulation of primordial follicle activation remains unclear [11]. In this study, we analyzed the ERβ-regulated genes in developing rat ovaries to understand the underlying mechanisms. We demonstrate that the expression of Indian hedgehog (*Ihh*) in neonatal rat ovaries is entirely ERβ dependent. 

During folliculogenesis, granulosa cells of activated follicles express hedgehog proteins, which are essential for the development, growth, and differentiation of theca cells [14,15,16,17,18]. The loss of hedgehog signaling or aberrant activation of hedgehog signaling disrupts theca cell differentiation, leading to failed ovulation [16,19]. The *Ihh* and desert hedgehog (*Dhh*) genes are expressed in granulosa cells, whereas the hedgehog receptor *Ptch1/Ptch2* and the signal transducer *Smo*, as well as hedgehog target genes like *Gli1/Gli2/Gli3* transcription factors, are expressed in theca cells [20,21]. It is also important that ERβ is the predominant estrogen receptor in granulosa cells, while ERα plays a dominant role in TCs [22,23]. However, whether ERβ plays any regulatory role in *Ihh* or *Dhh* gene expression in granulosa cells or ERα in the expression of hedgehog receptors or downstream signaling molecules in theca cells is unknown. 

## 2. Materials and Methods

### 2.1. Animal Model

*Erβ^KO^* Holtzman Sprague Dawley rats were included in this study. An *Erβ^KO^* rat model was generated by targeting the exon 3 in the *Erβ* gene, causing a frameshift and null mutation [8]. The rats were screened for the presence of mutations via genotyping PCR using tail-tip DNA samples as described previously [8]. All procedures were performed following the protocols approved by the University of Kansas Medical Center (KUMC) Animal Care and Use Committee.

### 2.2. Histological Evaluation of Ovarian Phenotypes

Ovaries were collected from 4.5-to-8.5-day-old *Erβ^KO^* and age-matched wild-type rats. One ovary from each rat was embedded in OCT (Fisher Scientific, Hampton, NH, USA), frozen immediately, and preserved at −80 °C until sectioning. The other ovary was snap-frozen in liquid nitrogen and preserved at −80 °C until processing for RNA extraction. Histological sections were prepared using a cryotome. The frozen sections were prepared from whole ovaries at a thickness of 6 µm and placed on charged glass slides (Fisher Scientific). The ovary sections were stained with hematoxylin and eosin following standard procedures [24]. The H&E-stained sections were thoroughly examined for follicle morphology and counted for follicles in each stage of development, as we have described previously [11]. Histological analyses were performed on one ovary from at least 3 independent wildtype rats or 3 independent *Erβ^KO^* rats.

### 2.3. Total Follicle Counting in the Serial Sections of the Whole Ovary 

The ovaries were collected and fixed in 4% formaldehyde overnight, processed, and embedded in paraffin following standard procedures [25,26,27]. Whole ovaries were serially sectioned at 6 µM thickness and stained with hematoxylin and eosin [25,26,27]. The stages of follicle development, including primordial, primary, secondary, early antral, and antral, were determined as described previously [28]. The primordial follicles were recognized as small oocytes surrounded by a few flattened granulosa cells, the primary follicles contained larger oocytes surrounded by a single layer of cuboidal granulosa cells, and the oocytes in the secondary follicles were surrounded by multiple layers of granulosa cells. The tertiary follicles were categorized into early antral and antral follicles based on the appearance and extent of the cavities within the follicles. Atretic follicles were recognized by pyknotic granulosa cells surrounding degenerated oocytes. The follicles were counted on every fifth section under light microscopy [25,26]. To avoid counting the same follicle more than once, primordial, and primary follicles were counted if they exhibited a nucleus, whereas the secondary, early antral, and antral follicles were counted only in the presence of a nuclei with prominent nucleoli [25,26,27]. The counts were multiplied by 5 to obtain the total count of follicles in the whole ovary. Ovarian follicle counting was performed on both ovaries collected from at least 3 independent wildtype rats or 3 independent *Erβ^KO^* rats. 

### 2.4. Detection of Differentially Expressed Genes 

Gene expression at the mRNA level was evaluated through RNA sequencing (RNA-Seq), and RT-qPCR analyses. Total RNA was extracted from the whole ovary with TRI Reagent (Millipore-Sigma, St. Louis, MO, USA), and RNA samples with an RIN value ≥ 9 were considered for the RNA-Seq library preparation. Next, 500 ng of total RNA from each sample was used for the RNA-Seq library preparation using the TruSeq Stranded mRNA kit (Illumina, San Diego, CA, USA) following the manufacturer’s instructions. The cDNA libraries were evaluated for quality at the KUMC Genomics Core and then sequenced on an Illumina HiSeq X sequencer at Novogene Corporation (Sacramento, CA, USA). All RNA-Seq data have been submitted to the Sequencing Read Archive (SRX6955095-6955104). RNA-Seq data were analyzed using CLC Genomics Workbench (Qiagen Bioinformatics, Redwood City, CA, USA) as described in our previous publications [9,10]. Gene ontology analysis of differentially expressed genes was performed using https://pantherdb.org, accessed on 30 November 2023 and the molecular functional pathways were determined. Ingenuity Pathway Analysis (IPA, Qiagen Bioinformatics) of the differentially expressed genes revealed functional pathways and different upstream or downstream signaling pathways.

### 2.5. Evaluation of Gene Expression Using RT-qPCR

Total RNA was extracted from the PND 8.5 ovaries using TRI Reagents (Sigma-Aldrich). Next, 1000 ng of total RNA from each sample was used to prepare cDNA using High-Capacity Reverse Transcription Kits (Applied Biosystems, Foster City, CA, USA). RT-qPCR amplification of cDNAs was carried out in a 10 µL reaction mixture containing Applied Biosystems Power SYBR Green PCR Master Mix (Thermo Fisher Scientific). Amplification and fluorescence detection of RT-qPCR was carried out on an Applied Biosystems QuantStudio Flex 7 Real-Time PCR System (Thermo Fisher Scientific). The ΔΔCT method was used to quantify the target mRNA expression level normalized to Rn18s (18S rRNA), as described in our previous publications [9,10]. For RT-qPCR analyses, any group of cDNAs at each time point included cDNAs prepared from 6 independent rat ovaries. 

### 2.6. RNAScope In Situ Hybridization

Ten µm thick frozen sections of postnatal day 8.5 and postnatal day 21.5 *Erβ^KO^* and age-matched wildtype rat ovaries were hybridized with RNAScope probes for rat *Ihh* (ACD Bio, Newark, CA, USA). In situ hybridization was performed on the frozen sections following the manufacturer’s instructions. After hybridization, the sections were reacted with HD brown and red assay reagents. The slides were mounted on Eco Mount (BioCare Medical, Pacheco, CA, USA), and images were captured using a Nikon 80i light microscope. In situ hybridization for each gene (mRNA) target was performed on ovary sections prepared from at least from three independent rats of each genotype (wildtype or *Erβ^KO^*). 

### 2.7. Statistical Analysis 

Each RNA-Seq library was prepared using pooled RNA samples from three individual wildtype or *Erβ^KO^* rats. Each group of RNA sequencing data consisted of three to four different libraries. For the RT-PCR experiments, each cDNA was prepared from pooled RNA from three rat ovaries of the same genotype. Either wildtype or the *Erβ^KO^* group consisted of 6 cDNAs. Follicle counting was performed on ovaries collected from at least three wildtype or *Erβ^KO^* rats. All of the laboratory investigations were repeated to ensure reproducibility. The data are presented as the mean ± standard error (SE). The results were analyzed using one-way ANOVA, and the significance of the mean differences was determined by Duncan’s post hoc test, with *p* ≤ 0.05. The statistical calculations were performed through the use of SPSS 22 (IBM, Armonk, NY, USA).

## 3. Results

### 3.1. Increased Primordial Follicle Growth Activation in Erβ^KO^ Rats

Upon histological examination, the activation of medullary follicles appeared to be accelerated in *Erβ^KO^* rat ovaries. However, the number of activated follicles in *Erβ^KO^* was not significantly different from that in wildtype ovaries on postnatal day 4.5 and postnatal day 6.5 (Figure 1). In contrast, there were remarkable increases in primordial follicle activation in postnatal day 8.5 *Erβ^KO^* ovaries (Figure 1E–G and Appendix A). On postnatal day 8.5, wildtype rat ovaries contained approximately 60% of the total follicles in the primordial stage, and approximately 30% were in the activated primary stage (Figure 1G). In contrast, on postnatal day 8.5 *Erβ^KO^* ovaries, the number of activated follicles was increased to approximately 50%, and the number of primordial follicles was reduced to 40% (Figure 1G). The number of healthy secondary follicles and atretic secondary follicles also increased in postnatal day 8.5 *Erβ^KO^* ovaries (Figure 1G). However, the total follicular counts of ovarian follicles in various stages of development remained similar in wildtype and *Erβ^KO^* ovaries.

### 3.2. ERβ Regulates Genes in Neonatal Rat Ovaries 

To identify the transcriptional targets of ERβ that are involved in regulating the rate of primordial follicle activation, we performed RNA-Seq analyses of postnatal day 4.5, 6.5, and 8.5 *Erβ^KO^* ovaries and age-matched wildtype rat ovaries. RNA-Seq analyses of postnatal day 8.5 ovaries detected 19,620 genes out of 29,541 genes in the rat reference library (mRatBN7.2.110). Of the 19,620 genes, 14,403 genes had TPM values ≥ 1 and 11,933 genes had TPM values ≥ 5. We observed that 642 genes were differentially expressed in *Erβ^KO^* postnatal day 8.5 ovaries (FDR *p*-value ≤ 0.05; absolute fold change 2, TPM ≥ 1). When we analyzed the high-copy genes (e.g., TPM ≥ 5), we found that 295 genes were differentially expressed (FDR *p*-value ≤ 0.05; absolute fold change 2, TPM ≥ 5). We observed that *Ihh*, an ovary-specific abundantly expressed gene, was one of the top downregulated genes in *Erβ^KO^* ovaries. It was also associated with similar downregulation of an IHH target gene (*Hhip*), and the differential expression data were reproducible through the use of RT-qPCR analyses (Table 1, Figure 2, Figure 3 and Appendix A). Therefore, for further analysis, we focused on differentially expressed genes that are related to hedgehog signaling, ovarian steroidogenesis, and regulators of ovarian folliculogenesis. Differential expression of these genes was detected in postnatal day 4.5 ovaries (Appendix A) and it became more distinct in day 6.5 (Appendix A) and 8.5 ovaries (Figure 2A–H).

Among the differentially expressed genes with TPM values ≥ 5, we also identified 10 transcriptional regulators as being downregulated and 8 transcriptional regulators as being upregulated (Table 2). The top 5 upregulated transcription factors were *Rbpjl*, *Dbx2*, *Dmrt1*, *Npas2*, and *Pou5f1*. In contrast many of the downregulated transcription factors, including *Pparg*, *Mycn*, *Fosl2*, *Egr1*, *Osr2*, *Nr5a1*, *Fox1*, *Fos*, *Myc*, and *Nr4a1*, are known for their role in regulating ovarian folliculogenesis [29,30,31,32,33]. 

Gene ontology analysis revealed that the differentially expressed genes in postnatal day 8.5 *Erβ^KO^* ovaries involved many molecular and functional pathways including ATP-dependent activity, antioxidant activity, molecular adaptor activity, transporter activity, transcription, and translation regulator activity (Appendix A). Ingenuity Pathway Analysis of the differentially expressed revealed different upstream and downstream signaling mechanisms such as *Esr1*, *Fgf2*, *Egf*, *Egfr*, *Vegf*, *Edn1*, *Agt*, *Csf2*, and *Il1β*, which are involved in the regulation of the development of vasculature or vasculogenic or angiogenesis, the growth of ovarian follicles, the proliferation of ovarian granulosa cells, and the growth of the whole ovary (Appendix A).

### 3.3. ERβ Regulates Indian Hedgehog Signaling

We observed a low level of mutant *Erβ* mRNA expression in postnatal day 4.5, 6.5, and 8.5 *Erβ^KO^* ovaries (Figure 3A). Although we did not detect any change in the expression of *Foxl2* (Figure 3B), we found that the loss of ERβ signaling disrupted *Ihh* expression in early neonatal ovaries (Figure 2C,D). Of the differentially expressed transcripts with TPM values ≥ 5.0, *Ihh* was one of most downregulated genes among the ovary-specific genes (Table 1, Figure 3C). Similar downregulation was also observed for the expression of IHH-target gene *Hhip* (Figure 3D). In addition to *Ihh* and *Hhip*, the expression of hedgehog receptor *Patch1* and the downstream transcriptional regulator *Gli1* was downregulated in *Erβ^KO^* ovaries (Figure 3E,F). Strikingly, the expression levels of *Gdf9* and *Bmp15*, which are known positive regulators of *Ihh* expression, were significantly higher in *Erβ^KO^* ovaries (Figure 3G,H). The RNA-Seq data were verified through the use of RT-qPCR, which showed similar results (Appendix A).

### 3.4. Altered Expression of Steroidogenic Enzyme Genes in Erβ^KO^ Ovaries

IHH signaling plays an important role in regulating the expression of steroidogenic enzymes. While analyzing the RNA-Seq data, we observed that the loss of ERβ disrupted the expression of *Hsd17b1* and *Cyp11a1* in *Erβ^KO^* rat ovaries (Figure 2E,F and Figure 4A,C). The expression of the FSH responsive gene, *Star,* and gonadotropin receptor, *Fshr*, remained unchanged in *Erβ^KO^* rat ovaries (Figure 4B,E). However, the expression of *Lhcgr* was significantly downregulated in postnatal day 8.5 *Erβ^KO^* rat ovaries (Figure 4F). The RNA-Seq results were further confirmed through the use of RT-qPCR, which showed comparable results (Appendix A). 

### 3.5. ERβ Regulates Ihh Only during the Early Neonatal Period

The downregulation of both *Ihh* and *Hhip* in *Erβ^KO^* rat ovaries was persistent, and this downregulation increased significantly from postnatal day 4.5 through to postnatal day 8.5 (*p* = 0.000) (Figure 3A,B). The data were confirmed through the use of RT-qPCR analysis (Figure 5A,C). Interestingly, the expression of both *Ihh* and *Hhip* was upregulated in postnatal day 16.5 *Erβ^KO^* rat ovaries and remained high, with it becoming similar to that of wildtype ovaries through to postnatal day 21.5 and postnatal day 28.5 (Figure 5B,D). To identify the cells that express *Ihh* in *Erβ^KO^* rat ovaries from postnatal day 16.5 onwards, we performed RNAScope in situ hybridization for rat *Ihh* using postnatal day 8.5 and postnatal day 21.5 *Erβ^KO^* and age-matched wildtype ovaries. We observed that *Ihh* mRNA is expressed only in the granulosa cells of activated follicles, mostly secondary and early antral follicles in postnatal day 8.5 wildtype ovaries (Figure 5E). No *Ihh* mRNA was detected in postnatal day 8.5 *Erβ^KO^* ovaries that contained the activated follicles of first wave origin (Figure 5F). As expected, both medullary and cortical follicles showed the expression of *Ihh* mRNA in postnatal day 21.5 wildtype ovaries (Figure 5G). However, there was no detection of *Ihh* mRNA in the medullary follicles of postnatal day 21.5 *Erβ^KO^* ovaries but strong *Ihh* signals in the activated large cortical follicles (Figure 5H).

### 3.6. ERβ-Regulated Ihh Impacts Theca Cell Development

IHH plays an important role in the development and differentiation of TCs in activated ovarian follicles. While ERβ is a major transcriptional regulator in granulosa cells, ERα is a crucial factor in theca cells. As the expression of IHH was affected in *Erβ^KO^* rat ovaries, we examined the expression of *Erα* mRNA in those (Figure 6). As expected, we detected no expression of *Erα* mRNA expression in postnatal day 8.5 *Erβ^KO^* rat ovaries (Figure 6B). In contrast, *Erα* mRNA was present in the postnatal day 21.5 *Erβ^KO^* rat ovaries, with it being a bit lower than in the wildtype follicles (Figure 6D). 

## 4. Discussion

Pregranulosa cells in primordial follicles do not express *Ihh*; it is expressed in the granulosa cells of activated ovarian follicles [17]. The loss of *Ihh* expression in neonatal rat ovary was associated with the loss of *Hhip* expression in a similar fashion to that reported in an *Ihh^KO^* mouse ovary [17]. Previous studies have suggested that GDF9 and BMP15 expressed by the oocytes in activated follicles induce the expression of *Ihh* in mouse ovaries [16,34]. In this study, we observed a very low level of *Ihh* and *Hhip* gene expression in neonatal *Erβ^KO^* rat ovaries despite high levels of *Gdf9* or *Bmp15* expression. The presence of GDF9 and BMP15 cannot induce *Ihh* expression in the absence of ERβ in neonatal rat ovaries. We demonstrate that signaling mediated through ERβ is essential for *Ihh* expression in the granulosa cells of the activated ovarian follicles. Such an important role of estrogen signaling in the regulation of *Ihh* expression is a novel finding, which may have a significant impact on future studies.

RNA-Seq analyses identified altered expression of several steroidogenic enzymes in *Erβ^KO^* rat ovaries, which can be either of granulosa cell or theca cell origin. It is well known that the loss of ERβ affects the expression of granulosa cell genes involved in steroidogenesis, including *Cyp11a1* and *Cyp19a1* [8,9]. However, in *Ihh^KO^* mouse ovaries, a similar impact on steroidogenic enzymes was reported [17]. Taken together, we may conclude that the defective expression of *Ihh* from granulosa cells may also affect steroidogenesis in theca cells. Steroidogenesis during early life does not have any role in reproductive development; however, it may have a crucial role in the regulation of folliculogenesis. 

We observed that the downregulation of *Ihh* expression in *Erβ^KO^* rat ovaries was prominent during the first week of the rat’s life, when the majority of the activated follicles are derived from the first wave of primordial follicles [2,3]. We observed that the differential expression disappeared in the second week of life, when the activated follicles started developing from the second wave of primordial follicles [2,3]. It has been shown that while the first wave of primordial follicles develop in the medullary region, the second wave of primordial follicles develop in the ovarian cortex [2,3]. It is also well established that the granulosa cells of the first wave of primordial follicles and the second wave of primordial follicles have a distinct origin [35]. The granulosa cells in the two waves of ovarian follicles thus express differentially regulated genes. While the pregranulosa cells that are present in the first wave of primordial follicles initially express high levels of *Foxl2,* [36,37], the pregranulosa cells in the second wave of primordial follicles express high levels of *Lgr5* [38,39,40]. Thus, it is not unlikely that the regulation of *Ihh* expression is different in the granulosa cells of two waves of activated follicles.

Our in situ hybridization data clearly indicate that the activated medullary follicles of first wave origin do not express *Ihh* even in the third week of life. Accordingly, we can conclude that the expression of *Ihh* is dependent on ERβ in the first wave of follicles but not in the second wave of follicles. Further understanding of such a differential regulation of *Ihh* expression may have an impact on extraovarian tissues where hedgehog signaling and estrogen signaling play an important role, e.g., bone formation or tumorigenesis. 

Granulosa cells possess a high level of ERβ expression [22,23]. Our findings suggest that *Ihh* expression in granulosa cells is regulated by this ligand activated transcription factor. Granulosa cells in the activated wildtype ovarian follicles express the *Ihh* gene, which in turn acts on premature theca cells for the induction of their development and differentiation [16]. We observed that the IHH target gene *Hhip*, which is of theca cell origin, is dramatically downregulated in *Erβ^KO^* rat ovaries. Thus, it is likely that one will observe the downregulation of the steroidogenic enzyme genes in theca cells, particularly those regulated by IHH signaling [18]. Our in situ hybridization data also suggest the defective development of theca cells in *Erβ^KO^* rat ovaries, particularly during the first week of life. Moreover, neonatal ovaries possess mechanisms to detect steroidogenesis and respond dynamically to regulate the process of primordial follicle formation and activation [41]. 

Although mice harboring ovary-specific *Ihh^KO^* did not exhibit any ovulatory dysfunction, this does not exclude a possible role of IHH signaling in regulating primordial follicle activation. Moreover, the representative images of *Ihh^KO^* mouse ovaries appeared smaller than that of wildtype mouse ovaries, comparable to the smaller sizes of *Erβ^KO^* mouse or *Erβ^KO^* rat ovaries [8]. We hypothesize that ERβ-regulated expression of *Ihh* from the activated follicles may act on dormant primordial follicles to control excessive primordial follicle activation, which has been observed in *Erβ^KO^* rat ovaries [11]. However, further studies are required to prove the mechanistic role of IHH in regulating primordial follicle activation.

## 5. Conclusions

It is widely accepted that *Ihh* is expressed in the granulosa cells of activated ovarian follicles. We observed that ERβ regulates the expression of *Ihh* in granulosa cells. However, such ERβ-dependent expression of *Ihh* was limited to the first wave of ovarian follicles. The expression of *Ihh* in the granulosa cells of second wave follicles did not require the presence of ERβ. Further studies are required to understand the significance of ERβ-regulated *Ihh* expression in the first wave of ovarian follicles.

## Figures and Tables

**Figure 1 cells-13-00644-f001:**
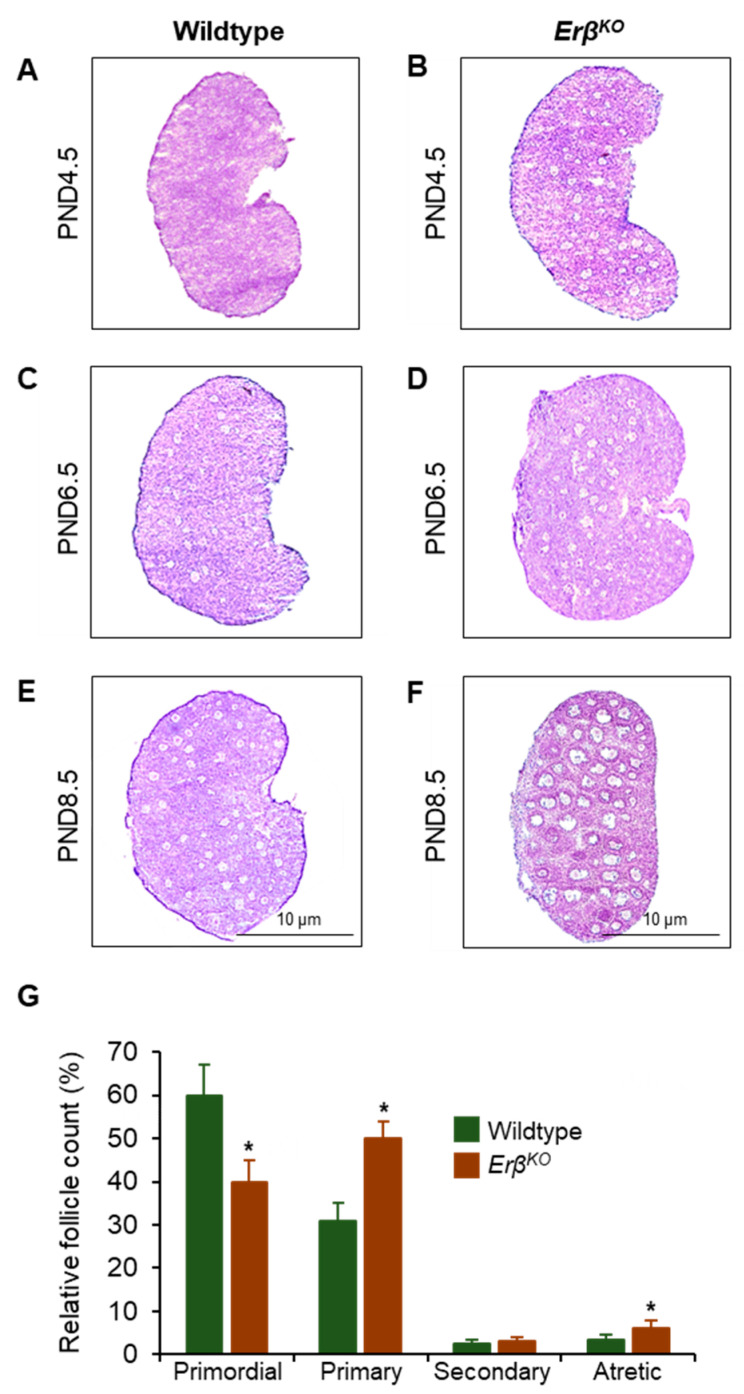
Loss of *Erβ^KO^* results in increased primordial follicle activation in immature rat ovaries. We detected increased growth of activated medullary follicles in *Erβ^KO^* rat ovaries compared to age-matched wildtype ovaries (**A**–**G**). An increased number of activated primary, secondary, and atretic follicles was evident in postnatal day 8.5 *Erβ^KO^* rat ovaries (**E**–**G**). While the wildtype rat ovaries contained more primordial follicles, they were replaced by primary follicles in *Erβ^KO^* rat ovaries (**E**–**G**). Data shown as the mean ± SE, * *p* < 0.05, *n* > 3.

**Figure 2 cells-13-00644-f002:**
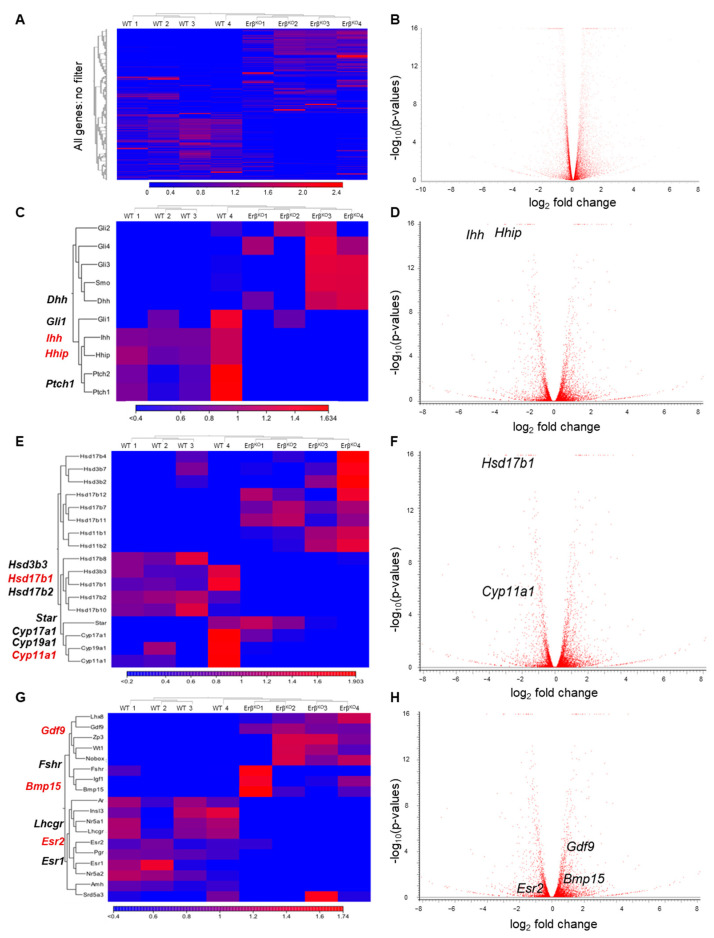
RNA-Seq analysis of postnatal day 8.5 rat ovaries. RNA-Seq analysis (heat maps in the left panel and volcano plots in the right panel) of postnatal day 8.5 rat ovaries showing the differential expression of genes in the whole ovary (no filter) (**A**,**B**), related to the hedgehog pathway (**C**,**D**), steroidogenesis (**E**,**F**), and the key genes involved in the process of folliculogenesis (**G**,**H**). Data are shown as CLC Genomic Workbench Analysis.

**Figure 3 cells-13-00644-f003:**
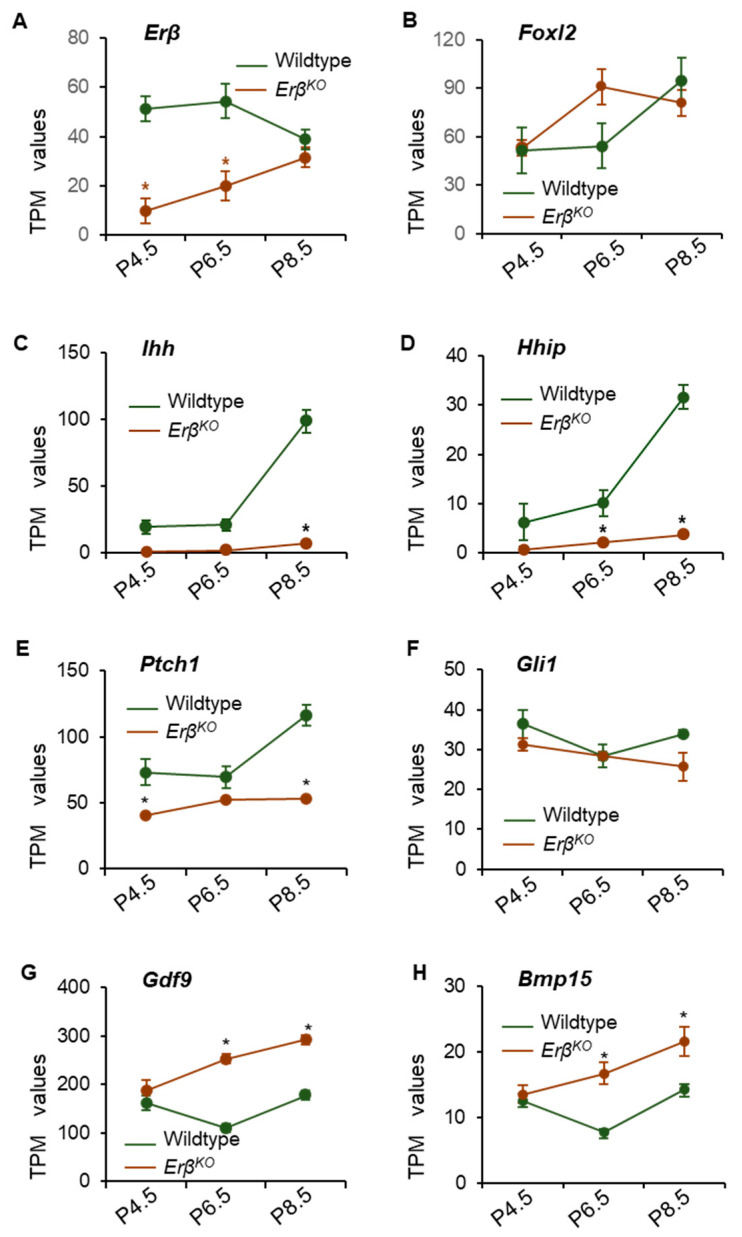
Loss of ERβ disrupts Indian hedgehog signaling in *Erβ^KO^* rat ovaries (**A**). RNA-Seq analysis of postnatal day 4.5, 6.5, and 8.5 showed downregulation of *Erβ* but not another grnaulosa cell-specific transcription factor *Foxl2* (**B**). We observed a marked downregulation of *Ihh*, and its downstream targets *Hhip* and *Gli1*, as well as its receptor *Ptch1* in *Erβ^KO^* ovaries (**C**–**F**). *Ihh* expression was remarkably low (**C**) despite its known regulators *Gdf9* (**G**) and *Bmp15* (**H**) being significantly upregulated in *Erβ^KO^* ovaries. Data shown as the mean ± SE TPM values; * *p* ≤ 0.05; *n* = 4 (RNA-Seq).

**Figure 4 cells-13-00644-f004:**
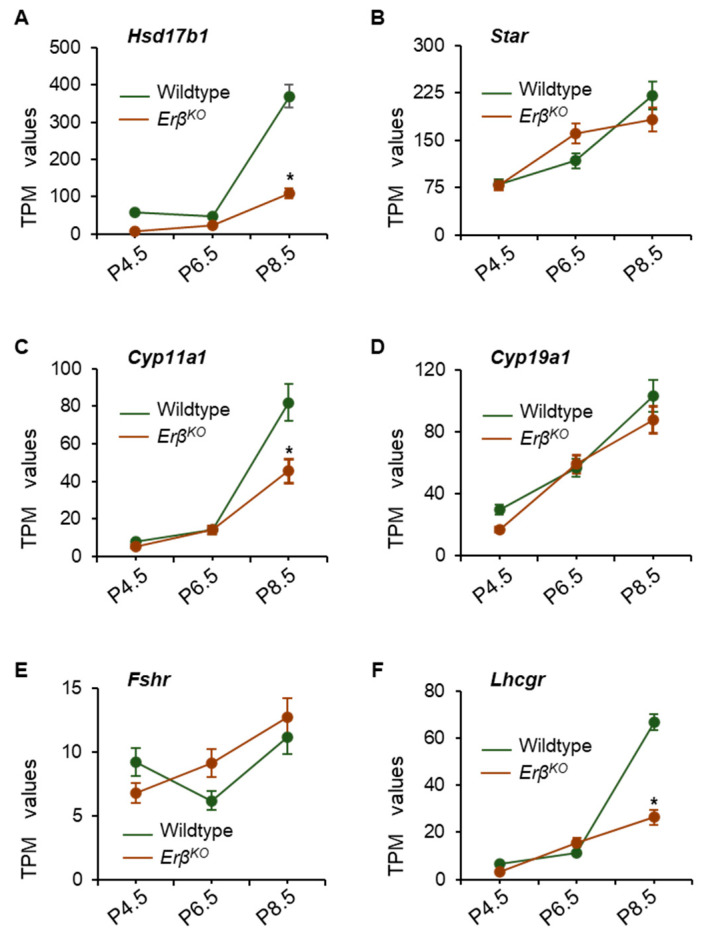
Loss of ERβ dysregulated the expression of steroidogenic enzymes in *Erβ^KO^* rat ovaries. RNA-Seq analysis of postnatal day (**P**) 4.5, 6.5, and 8.5 ovaries showed marked downregulation of *Hsd17b1*, and *Cyp11a1* (**A**,**C**) in *Erβ^KO^* ovaries. However, the expression of *Star* and *Cyp19a1* did not show any significant changes in *Erβ^KO^* ovaries (**B**,**D**). While the expression of *Fshr* remained unchanged, the expression of *Lhcgr* was significantly downregulated in postnatal day 8.5 *Erβ^KO^* ovaries (**E**,**F**). Data shown as the mean ± SE TPM values; * *p* ≤ 0.05; *n* = 4.

**Figure 5 cells-13-00644-f005:**
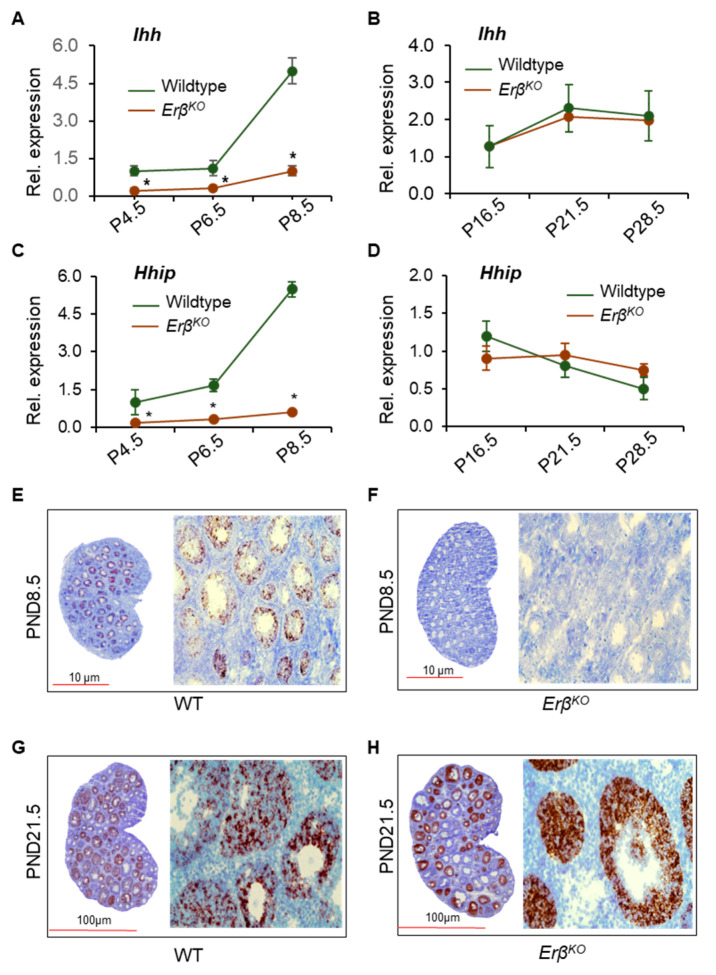
ERβ regulates *Ihh* gene expression only in the first wave of ovarian follicles. RT-qPCR analysis of postnatal day (P) 4.5, 6.5, and 8.5 ovaries showed increasing downregulation of *Ihh* and *Hhip* expression in *Erβ^KO^* ovaries (**A**,**C**). However, the downregulation of *Ihh* and *Hhip* expression disappeared in *Erβ^KO^* ovaries starting from postnatal day 16.5 onwards (**B**,**D**). RNAScope in situ hybridization detected the expression of *Ihh* mRNA in the activated medullary follicles in postnatal day 8.5 wildtype (WT) ovaries (**E**) but not in the age-matched *Erβ^KO^* ovaries (**F**). Granulosa cells in all activated follicles within the medullary as well as the cortical region of postnatal day 21.5 WT ovaries showed the expression of *Ihh* mRNA (**G**). In contrast, prominent expression of *Ihh* mRNA was detected only in the activated peripheral (cortical) follicles in PND 21.5 *Erβ^KO^* ovaries but not in the activated medullary follicles (**H**). RT-qPCR data are shown as relative expression according to mean ± SE values; * *p* ≤ 0.05; n = 6.

**Figure 6 cells-13-00644-f006:**
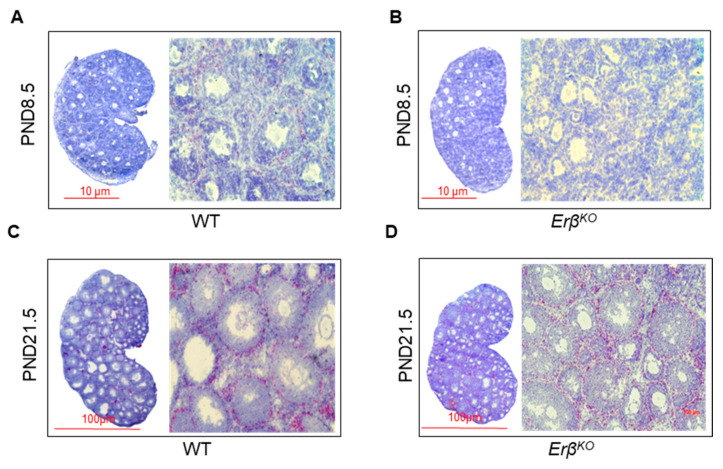
Loss of *Ihh* expression affected theca cell development in *Erβ^KO^* ovaries. In situ hybridization was performed for the detection of *Erα* mRNA (a marker of theca cells) in postnatal day 8.5 and postnatal day 21.5 wildtype (WT) and age-matched *Erβ^KO^* ovaries. The histology sections showed the presence of *Erα* mRNA in the peripheral regions surrounding the secondary follicles in postnatal day 8.5 wildtype ovaries (**A**). However, the postnatal day 8.5 *Erβ^KO^* ovaries did not show any expression of *Erα* mRNA (**B**). In contrast, *Erα* mRNA was abundantly expressed in postnatal day 21.5 *Erβ^KO^* ovaries, comparable to that of age-matched WT ovaries (**C**,**D**).

**Table 1 cells-13-00644-t001:** Top 10 differentially expressed genes in *Erβ^KO^* postnatal day 8.5 rat ovaries (absolute fold change ≥ 2, FDR *p*-value ≤ 0.05, and maximum TPM ≥ 5).

**1 A. Top 10 Downregulated Genes**						
**Name**	**Chrom**	**ENSEMBL**	**Region**	**Max TPM**	**Fold Change**	**FDR *p*-Value**
*ENSRNOG* *00000065112*	1	ENSRNOG00000065112	167210874..167213149	7.4	−902.74	0.00
*Lce1m*	2	ENSRNOG00000009581	complement (178637495..178638697)	6.26	−280.11	0.00
*ENSRNOG* *00000065518*	2	ENSRNOG00000065518	complement (178666387..178668404)	6.7	−149.27	0.00
*Abcb1b*	4	ENSRNOG00000066042	complement (25242798..25325199)	24.2	−28.71	0.00
*Snrpc-ps3*	10	ENSRNOG00000017586	complement (14191662..14192401)	6.69	−16.78	0.00
*Ihh*	9	ENSRNOG00000018059	complement (76504315..76510532)	118.52	−15.20	0.00
*Fam111a*	1	ENSRNOG00000012067	209640953..209656547	156.28	−8.39	0.00
*Hhip*	19	ENSRNOG00000018268	27863213..27952528	39.48	−7.87	0.00
*Gal*	1	ENSRNOG00000015156	complement (200650439..200654959)	9.89	−6.63	0.00
*Kcns2*	7	ENSRNOG00000066111	66022352..66028422	9.77	−6.52	0.00
**1 B. Top 10 Upregulated Genes**						
**Name**	**Chrom**	**ENSEMBL**	**Region**	**Max TPM**	**Fold Change**	**FDR *p*-Value**
*LOC685716*	11	ENSRNOG00000054404	55280418..55303827	10.84	315.03	0.00
*LOC100360791*	1	ENSRNOG00000033517	19107386..19108232	470.98	17.11	0.00
*Slc27a5*	1	ENSRNOG00000019626	complement (73616564..73627172)	5.13	13.92	0.00
*Aldh1a7*	1	ENSRNOG00000017878	complement (218201472..218248906)	9.85	11.66	0.00
*ENSRNOG* *00000066616*	3	ENSRNOG00000066616	149220737..149228279	5.05	9.67	0.00
*Actc1*	3	ENSRNOG00000008536	complement (100811987..100817523)	7.04	9.55	0.00
*Adcyap1r1*	4	ENSRNOG00000012098	84593892..84642700	25.86	9.29	0.00
*Adh6_1*	2	ENSRNOG00000012436	226797303..226808892	13.71	6.90	0.00
*Gabrr3*	11	ENSRNOG00000001679	complement (40902812..40955263)	10.62	6.76	0.00
*Cpa2*	4	ENSRNOG00000028092	59160357..59183674	250.63	6.74	0.00

**Table 2 cells-13-00644-t002:** Differentially expressed transcription factors in *Erβ^KO^* postnatal day 8.5 rat ovaries.

Name	Chrom	ENSEMBL	Region	Max TPM	Fold Change	FDR *p*-Value
*Rbpjl*	3	ENSRNOG00000026295	153134140..153146513	10.90	5.79	0.00
*Dbx2*	7	ENSRNOG00000006885	complement (126772227..126802564)	5.18	5.36	0.00
*Dmrt1*	1	ENSRNOG00000016075	223142859..223241333	6.22	5.13	0.00
*Npas2*	9	ENSRNOG00000013408	41463830..41642320	36.43	4.07	0.00
*Pou5f1*	20	ENSRNOG00000046487	complement (3223129..3227891)	20.60	2.59	0.00
*Setbp1*	18	ENSRNOG00000016208	complement (72191035..72552556)	5.24	2.18	0.00
*Hoxc6*	7	ENSRNOG00000063956	134135502..134148392	6.50	2.07	0.00
*Scx*	7	ENSRNOG00000021812	108176608..108178626	13.02	2.04	0.00
*Pparg*	4	ENSRNOG00000008839	148423194..148548468	14.06	−2.04	0.00
*Mycn*	6	ENSRNOG00000051372	complement (35717764..35723590)	58.93	−2.05	0.00
*Fosl2*	6	ENSRNOG00000068412	complement (24300956..24320034)	36.83	−2.10	0.00
*Egr1*	18	ENSRNOG00000019422	26462981..26466766	67.29	−2.17	0.00
*Osr2*	7	ENSRNOG00000011136	66487839..66495224	92.84	−2.26	0.00
*Nr5a1*	3	ENSRNOG00000012682	complement (22465502..22486328)	173.03	−2.35	0.00
*Foxo1*	2	ENSRNOG00000013397	136312168..136387790	97.01	−2.38	0.00
*Fos*	6	ENSRNOG00000008015	105121170..105124036	23.72	−2.52	0.00
*Myc*	7	ENSRNOG00000004500	93593705..93598630	47.62	−2.60	0.00
*Nr4a1*	7	ENSRNOG00000007607	132374840..132389297	87.54	−4.97	0.00

## Data Availability

All RNA-Seq data have been submitted to the Sequencing Read Archive (SRX6955095-6955104).

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
