# Peer review of "ERβ Regulation of Indian Hedgehog Expression in the First Wave of Ovarian Follicles"

_cells, 2024, doi:10.3390/cells13070644_

Round 1

Reviewer 1 Report

Comments and Suggestions for Authors

The manuscript delves into the intricate regulatory role of ERb in granulosa cells concerning Ihh and Dhh, along with their receptors in the theca cells, as part of a comprehensive exploration of the inhibitory mechanisms orchestrated by ERb in primordial follicle activation. The authors present compelling data demonstrating that the loss of ERb expression results in diminished Ihh and its downstream genes in granulosa cells, coupled with reduced Patch 1 and its downstream gene in the theca cells. The study elucidates the functional significance of these altered gene expressions in the intricate processes of primordial follicle activation and theca cell development.

The specific aim of the research is to discern alterations in ERb-regulated genes linked to primordial follicle growth activation, shedding light on the heightened primordial follicle activation observed in ERbKO neonatal rats. Given the pivotal role of primordial follicle activation in establishing the ovarian reserve in women, the manuscript offers innovative evidence of ERb's regulation of Ihh expression in the neonatal rat ovary, particularly in the granulosa cells of activated follicles. Despite the initial downregulation of Ihh and its regulated genes, subsequent recovery of Ihh expression implies that ERb predominantly regulates Ihh in the first wave of follicles in rat ovaries and not in later developmental stages.

While the manuscript is well-crafted, certain aspects warrant clarification and further exploration. The following questions and comments are presented for consideration:

1. The manuscript mentions that ERb-regulated Ihh impacts theca cell development, but additional details on the abnormalities in the theca cells are needed. Have you collected data illustrating abnormal theca cell development in this rat model, and how did you ascertain that the impact on theca cells is through Ihh-regulated genes and not other ERb-regulated genes?

2. The impact of ERbKO on Ihh expression is evident, but is there any other evidence of Ihh being directly regulated by ERb beyond gene expression data? Direct experimental evidence would strengthen this crucial point.

3. In Figure 3, showcasing RNAseq data for three time points, adding a graph illustrating ERb expression in the WT ovary would serve as a useful reference to visualize normal expression levels in relation to the presented gene changes.

4. Clarify the number of samples analyzed for Figure 1, both in terms of histological images (A-F) and quantitative data (G-H). In Figure 1G-H, consider adding an indication of significant results to clarify differences between ERbKO and WT. 

5. In the heat maps of Figure 2, consider changing the axis titles for sample names, particularly the designation "MT," which is unclear and not defined in the text.

6. A typographical error in line 293 refers to Figure 5D, stating that no Erβ mRNA was detected; it should read no Ihh mRNA was detected.

6. Ensure consistency in terminology, such as writing PDFs instead of PdFs, and maintain uniformity in gene and protein nomenclature throughout the manuscript.

7. Define the criteria used to categorize different follicle types and specify the rationale for distinguishing cortex and medulla areas in the manuscript.

8. Considering the low expression of Ihh in PND 6.5 or earlier ovaries, an in vitro ovarian culture with Ihh could complement the RT-qPCR and RNA-Seq data to observe changes in downstream gene expression.

9. The manuscript does not address why Ihh expression increases in the second wave follicles of PND 21.5 ovaries. Given the clear separation of first and second wave follicles, an explanation or discussion of this phenomenon is warranted.

Author Response

We are thankful for the reviewer's insightful comments. We have answered the reviewer’s queries, included the corrections in the revised manuscript, and included additional detailed explanations. We acknowledge that the reviewer’s suggestions have greatly improved our manuscript, which can be considered for publication in Cells. 

Reviewer 2 Report

Comments and Suggestions for Authors

The manuscript authored by Chakravarthi and colleagues describe the determination of the effects of ERß KO on the process of activation of primordial follicles in the rat ovary.  A striking observation was that the rate of entry of primordial follicles into the growing pool was increased when ERß was deleted. RNAseq was employed to determine the gene products dysregulated in the knockout, and the manuscript then focuses on Indian Hedgehog and its associated genes. The manuscript has, for the most part, been well prepared, but there are some clear deficiencies.  The figures are very well presented and didactic, and, in particular, the localization figures are convincing. The temporal effect, i.e. that the phenotype is present only in early postnatal development are interesting,  The following specific points were noted: 

1.  The overuse of acronyms renders the manuscript difficult to read.  

2.  It  is not clear why this gene was chosen as the focus of the study.

3.  The major finding is that the medullary follicles require ERß for Ihh signalling.  Nonetheless, the occurrence of ERß in this population of follicles has not been demonstrated.  Further, it is not clear how the effects on the medullary population of follicles are visited on the cortical follicles.  Figure 3F seems to demonstrate that the precocious activation is not restricted to to medullary population of follicles. 

4.  The experimental design is not clear, how many animals were in the RNAseq trial?  There is no information about collection of samples after PND 8.5,in the methods, but these data appear in the results (Figure 5B). 

5.  How were the early and late follicle populations counted?  Was their occurrence marked by their location in the ovary only? Did these follicles express Foxl2?

6.  Figure1F shows antral follicles, surprising at day 8.5.  Were these enumerated? 

7.  How was apoptosis established in the follicle counting scheme? 

8.  How many DEGs were revealed by RNAseq?  A gene ontology analysis would have provided considerably more data, particularly about genes up and downstream of Ihh. 

Author Response

We are thankful to the reviewer for her/his insightful comments. We have answered the reviewer’s queries, included the corrections in the revised manuscript, and included additional detailed explanations. We acknowledge that the reviewer’s suggestions have greatly improved our manuscript, which can be considered for publication in Cells. 

Reviewer 3 Report

Comments and Suggestions for Authors

The manuscript by Chakravarthi, “ERβ regulation of Indian hedgehog expression in the first wave  of ovarian follicles,” describes a study to determine effects of loss of ER-beta (ERb) in the perinatal ovary. These authors have previously demonstrated that loss of ERb leads to accelerated loss of the primordial follicle pool, via increased rate of primordial follicle activation. The major finding here is that loss of ERb leads to dysregulation of Ihh signaling in the first wave, but not subsequent waves, of ovarian follicles. The study is interesting, particularly in the attempt to evaluate effects on the first vs second wave of follicular activation, but there are a number of problems with the manuscript. First, it is absolutely essential that the authors report somewhere (either in the results or in the methods) the cutoffs used in RNAseq analysis (padj or q value and fold change). It is also absolutely essential that they use a false discovery rate cutoff and report this. Second, the manuscript has a number of errors and typos. Some of these (figure references) are listed below in the line-by-line comments, but others are not, and the entire manuscript should be read for correctness. Third, there are many acronyms in this manuscript and they make it more difficult to follow the story. Finally, there is no mechanistic information about the functional role of Ihh in the ovary. This would greatly strengthen the conclusions drawn. Without information about the mechanistic role of Ihh in the ovary, the paper is of minimal value, as it is not clear how exit of follicles from the ovarian reserve is being regulated. 

Line by line comments follow.

In section 2.2, if space allows, more detail is needed about quantification of follicles.

In section 2.6, it would be good to add the number of biological replicates for each experiment (including the follicle enumeration experiment), as this is not entirely clear.

In section 3.1, it would be preferable to report follicle counts as follicles / ovary rather than % of a total. In addition, it is absolutely essential to present the data as bar graphs, with error bars, rather than as pie charts in which variation in the population cannot be assessed by the reader. Finally, given that medullary follicle activation appears to be increased in the PND4.5 and PND6.5 ovaries, it would be interesting to draw a border between the medulla and cortex and assess and quantify these two follicular populations separately.

In section 3.2, no details are reported that allow overall assessment of the quality of the RNAseq data. These would include total number of DEG, cutoffs used to identify DEGs, PCA of overall clustering of samples, heatmaps of top DEG or all DEG, etc. Indeed, in figure 2, only PND8 data are shown and there is no mention of the PND6 and PND4 data, which would also be interesting and relevant here. In addition, there is no gene ontology or pathways analysis presented, which would be helpful in assessing overall changes in the ovary in response to loss of ERb and in understanding why Ihh signaling was chosen to pursue. Overall information about the RNAseq (including all three times assessed) needs to be included either as a figure or as a part of a figure or a supplemental figure.

In section 3.2, it is not clear why IHH signaling was chosen as a major focus.

Section 3.3 would be strengthened if the connection between GDF9/BMP5, ERb, and hedgehog could be made.

Line 182-183: the statement is made that Lhcgr does not change in ERbKO ovaries, but in figure 4f, it appears that Lhcgr is downregulated in ERbKO ovaries. In addition, the statement is made that Fshr is downregulated, but it appears to remain unchanged (figure 4e).

Line 211: what is the temporal statistical assessment from which the p value reported here comes?

Line 212-215, the authors refer to Ihh and Hhip, but only Ihh is shown in figure 5b. In addition, the authors refer upregulation of these molecules, but no upregulation is evident in figure 5b.

Section 3.5: why show Ihh and Hhip qPCR results both in figure 5 and in suppl figure 1? Are these two different qPCRs? Or the same?

Line 219-220: the authors state that no ERb mRNA was detected in ERbKO ovaries (figure 5D) but in the figure legend, the authors indicate that figure 5D shows Ihh expression.

Line 224: what is Figure F?

Figure 6: There are well-validated protocols for dual staining in RNAscope. It would be very interesting and support the authors’ conclusion to co-label for Ihh and ERa in day 21.5 ovaries and determine if the Ihh negative medullary follicles in the ERbKO ovaries have fewer ERa positive cells surrounding them, or if ERa expression is less strong in these cells.

Comments on the Quality of English Language

The paper has a number of typos and needs to be carefully read through for correctness. 

Author Response

(The authors gave the same response as above.)

Round 2

Reviewer 2 Report

Comments and Suggestions for Authors

The authors have successfully addressed my concerns with the earlier version of the manuscript. 

Author Response

Thank you for your kind evaluation.

Reviewer 3 Report

Comments and Suggestions for Authors

The authors have not addressed all of my concerns. 

1. Of primary concern, there is no mechanistic information about the functional role of Ihh in the ovary. This would greatly strengthen the conclusions drawn. Without information about the mechanistic role of Ihh in the ovary, the paper is of minimal value, as it is not clear how exit of follicles from the ovarian reserve is being regulated. As-is, the authors have essentially evaluated expression of Ihh family members and this paper is purely descriptive and not mechanistic. Cells is a good enough journal that some mechanistic information should be included. 

2. In response to query 3, it sounds as if the authors are reporting data that has already been published (follicle counts on PND8.5). Why report these results in the present paper if they have already been published? Why not simply reference the previous paper? If indeed these are new data, adding bar charts with error bars in addition to the pie graphs, which lack error bars or any indication of statistical significance, is important for the readers assessment of the quality of the result. 

3. The authors had reported a general summary of info about their RNAseq on PND8.5, but not PND6.5 or PND4.5. Was RNAseq run on these days? why is it not reported? 

4. in Line 212-215 in the original version and line 340-343 in the revised version, the authors refer to Ihh and Hhip, but only Ihh is shown in figure 5b. In addition, the authors refer upregulation of these molecules, but no upregulation is evident in figure 5b. ("Interestingly, the expression of both Ihh and Hhip was upregulated in postnatal day 16.5 ErβKO rat ovaries and remained high, which became similar to that of wildtype ovaries through postnatal day and postnatal day 28.5 (Figure 5B)." In response, the authors state that these are findings from the literature, but no literature is cited, and a figure is referenced that contains neither information about upregulation nor information about Hhip. 

Comments on the Quality of English Language

Typos and errors remain and the manuscript should be read carefully. 

Author Response

We are thankful to the reviewer for her/ her kind comments. We have addressed the individual comments in the following section.

Query1. Of primary concern, there is no mechanistic information about the functional role of Ihh in the ovary. This would greatly strengthen the conclusions drawn. Without information about the mechanistic role of Ihh in the ovary, the paper is of minimal value, as it is not clear how exit of follicles from the ovarian reserve is being regulated. As-is, the authors have essentially evaluated expression of Ihh family members, and this paper is purely descriptive and not mechanistic. Cells is a good enough journal that some mechanistic information should be included. 

Response 1. Please note that the primary focus of this manuscript is not the mechanisms underlying IHH-mediated regulation of primordial follicle activation. The manuscript reports the novel observation regarding ERβ regulation of Indian hedgehog expression in the first wave of ovarian follicles.

Our results indicate that ERβ-regulates the expression of Ihh in the first wave of activated follicles in neonatal ovaries, and IHH, in turn, regulates the expression of genes involved in steroidogenesis. Our data also suggest that in granulosa cells, Ihh expression can be regulated in either ERβ-signaling-dependent (in the first wave of follicles) or ERβ-signaling-independent ways (in the second wave of follicles).   

We have made comments regarding those in the Discussion section:

We observed that the IHH target gene Hhip, which is of theca cell origin, is dramatically downregulated in ErβKO rat ovaries. Thus, it is likely to observe the downregulation of the steroidogenic enzyme genes in theca cells, particularly those regulated by IHH signaling [18]. Our in-situ hybridization data also suggest a defective development of theca cells in ErβKO rat ovaries, particularly during the first week of life. Moreover, neonatal ovaries possess mechanisms to detect steroidogenesis and respond dynamically to regulate the process of primordial follicle formation and activation [41].

Query 2. In response to query 3, it sounds as if the authors are reporting data that has already been published (follicle counts on PND8.5). Why report these results in the present paper if they have already been published? Why not simply reference the previous paper? If indeed these are new data, adding bar charts with error bars in addition to the pie graphs, which lack error bars or any indication of statistical significance, is important for the reader's assessment of the quality of the result. 

Response 2. We agree with the reviewer’s suggestion. Accordingly, we have removed the pie charts (Figure 1 G and H) and included bar graphs (Figure 1 G and H revised).

Query 3. The authors had reported a general summary of info about their RNAseq on PND8.5, but not PND6.5 or PND4.5. Was RNAseq run on these days? why is it not reported? 

Response 3. We agree with the reviewer’s suggestion. Accordingly, we have included the summary of PND6.5 or PND4.5 RNA-sequencing data (Supplementary Figure 3 and 4).

Query 4. in Line 212-215 in the original version and line 340-343 in the revised version, the authors refer to Ihh and Hhip, but only Ihh is shown in figure 5b. In addition, the authors refer upregulation of these molecules, but no upregulation is evident in figure 5b. ("Interestingly, the expression of both Ihh and Hhip was upregulated in postnatal day 16.5 ErβKO rat ovaries and remained high, which became similar to that of wildtype ovaries through postnatal day and postnatal day 28.5 (Figure 5B)." In response, the authors state that these are findings from the literature, but no literature is cited, and a figure is referenced that contains neither information about upregulation nor information about Hhip. 

Response 4. We are thankful to the reviewer. We have included the Hhip expression data in our revised manuscript.

The upregulation of Ihh or Hhip expression at postnatal days 16.5, 21.5, and 28.5 in ErβKO rat ovaries are explained in the following statement:

Expression of Ihh or Hhip was downregulated in ErβKO rat ovaries at postnatal days 4.5, 6.5, and 8.5 compared to those in age-matched wildtype ovaries. Such downregulation of Ihh or Hhip expression in ErβKO rat ovaries was no longer observed at postnatal days 16.5, 21.5, and 28.5. It happened due to the upregulation of Ihh or Hhip expression in ErβKO rat ovaries at postnatal days 16.5, 21.5, and 28.5. Our statements do not indicate the expression in ErβKO rat ovaries was upregulated compared to wild-type ovaries.

The expression of Hhip is completely dependent on Ihh's expression (1-3). That has been shown in the IhhKO mouse model (2). Therefore, we detected the expression of Ihh in ovary sections. The expression of Ihh and Hhip is induced in activated follicles, those genes are not expressed in primordial follicles (1-3). These are established findings according to previous publications (1-3).

Query 5.  Comments on the Quality of English Language, Typos and errors remain, and the manuscript should be read carefully. 

Response 5. We are thankful to the reviewer. We have corrected the errors.

Reference (Query 4):

  1. Liu, C.; Rodriguez, K.F.; Brown, P.R.; Yao, H.H. Reproductive, Physiological, and Molecular Outcomes in Female Mice Deficient in Dhh and Ihh. Endocrinology 2018, 159, 2563-2575, doi:10.1210/en.2018-00095.
  2. Liu, C.; Peng, J.; Matzuk, M.M.; Yao, H.H. Lineage specification of ovarian theca cells requires multicellular interactions via oocyte and granulosa cells. Nat Commun 2015, 6, 6934, doi:10.1038/ncomms7934.
  3. Huang, C.C.; Yao, H.H. Diverse functions of Hedgehog signaling in formation and physiology of steroidogenic organs. Mol Reprod Dev 2010, 77, 489-496, doi:10.1002/mrd.21174.

Round 3

Reviewer 3 Report

Comments and Suggestions for Authors

The author have addressed my concerns. The only concern that remains is that the new supplemental figures showing the RNAseq data are very hard to read, with labeling that cannot be read at all. 

Author Response

We are thankful to the reviewer for her/ his expert suggestion. We have replaced the Supplementary Figures.